# Forest Fuel Bed Variation in Tropical Coastal Freshwater Forested Wetlands Disturbed by Fire

**Romeo de Jesús Barrios-Calderón** [1],*, **Dulce Infante Mata** [2],*, **José Germán Flores Garnica** [3] **and Jony R. Torres** [4]

1 Faculty of Agricultural Sciences, Autonomous University of Chiapas, Junction with Coastal Road and Huehuetán Station, Tuxtla Gutiérrez 30660, Chiapas, Mexico

2 Sustainable Management of Basins and Coast Zones, El Colegio de la Frontera Sur, Unit Tapachula, Highway Old Airport, km 2.5, Tapachula 30700, Chiapas, Mexico

3 National Institute of Forestry, Agricultural and Livestock Research (INIFAP), Pacific Regional Research Center, Experimental Field Centro-Altos of Jalisco, Los Colomos Park s/n, col. Providencia, Guadalajara 44660, Jalisco, Mexico; flores.german@gmail.com

4 I.T. del Valle del Yaqui, Academy of Biology, Coastal Zone Ecology Laboratory, National Technological Institute of Mexico, Avenue Tecnológico Block 611, Bácum 85276, Sonora, Mexico; jtorres.velazquez@itvy.edu.mx

* Correspondence: romeo.barrios@unach.mx (R.d.J.B.-C.); dulce.infante@gmail.com (D.I.M.); Tel.: +52-9622342421 (R.d.J.B.-C.); +52-9621840306 (D.I.M.)

**Abstract:** Tropical coastal freshwater forested wetlands in coastal regions are rapidly disappearing as a result of various disturbance agents, mainly wildfires caused by high accumulations of forest fuels. The objective of this study was to characterize the structure and composition of fuel beds in tropical coastal freshwater forested wetlands with three levels of disturbance at El Castaño, La Encrucijada Biosphere Reserve. Seventeen sampling units were used to describe the structure of the forest's fuel beds (canopy, sub-canopy, and understory). Fallen woody material and litter (surface and fermented) were characterized using the planar intersection technique. Diversity comprised eight species of trees, two shrubs, five lianas, and two herbaceous species. The vertical strata were dominated by trees between 2 and 22 m in height. The horizontal structure had a higher percentage of trees with normal diameter between 2.5 and 7.5 cm (61.4%) of the total. Sites with low disturbance had the highest arboreal density (2686 ind. ha$^{-1}$). Diversity of species showed that the Fisher, Margalef, Shannon, and Simpson $\alpha$ indices were higher in the low disturbance sites. The Berger–Parker index exhibited greater dominance in the sites with high disturbance. *Pachira aquatica* Aubl. Showed the highest importance value index and was the largest contributor to fuel beds. Sites with the highest disturbance had the highest dead fuel load (222.18 $\pm$ 33.62 Mg ha$^{-1}$), with woody fuels of classes 1, 10, and 1000 h (rotten) being the most representative. This study contributes to defining areas prone to fire in these ecosystems and designing prevention strategies.

**Keywords:** tropical coastal wetland; *Pachira aquatica*; flooded forest; forest fuel; forest fire

## 1. Introduction

In tropical regions, tropical coastal freshwater forested wetlands (TCFFWs) are regularly found close to mangrove forests in swampy terrain, with gentle slopes near the river bank, where freshwater has the most influence (via flood pulses) [1–3]. These plant communities group a large number of trees, shrubs, and lianas [4]. Since they develop in a transition zone, they play an important role in nutrient and biogeochemical processes in addition to other ecosystem services [5]. TCFFWs function as natural filters for pollutants and as refuges for wildlife [2]. In addition, they are an important carbon sink, and even store greater amounts of carbon in the soil than mangroves [4]. However, TCFFWs are an understudied tropical ecosystem.

TCFFWs are located in freshwater-influenced areas where one or two species commonly dominate [2,6]. One of these species is *Pachira aquatica* Aubl. (Common name: water

zapote), a floating tree that forms a false forest floor with produced and retained organic matter [7]. TCFFWs dominated by *P. aquatica* only share their space with certain flood-tolerant species [8]. In Mexico and Mesoamerica, the only TCFFWs dominated by *P. aquatica* are those distributed in Huimanguillo, Tabasco [9], Ciénaga del Fuerte, Laguna Chica, El Apompal in Veracruz [10], and La Encrucijada in Chiapas [11,12]. In La Encrucijada, the TCFFW of *P. aquatica* becomes associated with mangrove species [4]; however, human influence has caused the gradual disappearance of these ecosystems.

Climate change is affecting coastal wetlands around the world, causing changes in the structure, composition and function of species, loss of biomass, and an increase in forest fires. These impacts are aggravated by climatic factors (hurricanes, cyclones, etc.) and non-climatic factors (conversion of land for other uses, opening of gaps in the vegetation, burning to clear land, road and infrastructure development, timber extraction, and other human activities [13]. One of the disturbance threats leading to the loss of plant communities in TCFFW is forest wildfires, either from natural or anthropogenic causes. These play an important role in vegetation dynamics and land-use change [14–16]. Therefore, these coastal wetlands are considered vulnerable ecosystems that are threatened by the frequency and magnitude of fires [17–19]. Although TCFFWs remain flooded most of the year, they are highly productive and store woody forest fuels (i.e., branches, twigs, logs, fallen trees) and layers of litterfall. Leaf litter is found at different levels of decomposition on the soil surface [20]. In dry seasons, litter can lead to combustion processes; when the water level fades, the soil dries out, and all accumulated organic matter can burn [21,22]. This explains why this ecosystem presents recurrent fires, largely managed by the amount of woody fuel and litterfall (total accumulated biomass) and its quality (dry weight of biomass available for burning).

Forest fuels are the energy source that, in combination with topography, climate, and source of ignition, can cause fires that vary in magnitude and spread [23]. Since fires are an important environmental driver of ecosystem processes and biogeochemical cycles, studies that characterize and estimate forest fuels are of great relevance for fire management and prediction [24]. Fire behavior, as well as its potential impact, is defined by the available fuels, which vary in quantity and quality; therefore, in order to make comparisons between different ecosystems and conditions, the concept of fuel beds is used. Within these beds, fuels are characterized and quantified in six strata: canopy, shrubs, non-woody fuels, woody fuels, litter-lichen-mush, and ground fuels [25] in a given ecosystem. Thus, forest fuel beds characterize all combustible material, in relation to their spatial distribution (vertical and horizontal), and their ignition potential [26].

Fuel bed characterization integrates the analysis of the quantity and quality of forest fuel, considering both vertical and horizontal spatial distributions [27]. To characterize fuel beds, both live [26,28] and dead [29] fuels are considered in different categories or strata [30], among other factors that have caused the fragmentation of these forest ecosystems. Additional information to be considered is the frequency of fires due to anthropogenic activities, such as illegal wood extraction, the use of fire for poaching, and the proximity of the population center. Particularly in areas such as La Encrucijada Biosphere Reserve (LEBRE), Chiapas, these disturbing agents are a threat to ecosystems, such as TCFFW. CONANP [31] and CONAFOR [32] have collected information on fires in LEBRE since 1998 and have observed at least 20 fires in El Castaño, which have affected freshwater marshes (1200 ha), mangroves (800 ha), and *P. aquatica* TCFFW (450 ha). Therefore, the characterization of fuel beds and the quantification of fuel loads are two fundamental aspects to define strategies for the protection [33], particularly in TCFFW, being primary issues for the ecology of fire in coastal wetlands.

The study of fuel beds has been ©d wo©ldwide in different ecosystems [25,26,33–37], temperate [38–41] and tropical [42–45]. However, few studies have focused on exploring TCFFWs with different levels of disturbance. From this approach, diversity (species richness) can become a determining factor in the high or low accumulation of woody and litter fuels. Despite the relevance of fuel beds in the frequency and recurrence of fires in

coastal wetlands, particularly in TCFFW, there is insufficient information on the quantity, availability, continuity, and distribution of forest fuels in coastal wetlands in Mexico. Although TCFFWs present different levels of disturbance, it has been noted that the presence of the species *P. aquatica* may have an important relationship with the high combustibility; therefore, it is expected that in areas where this species has a high value of importance, there is a greater accumulation of dead fuels, while in areas where species richness increases, the accumulation of dead fuels is lower.

There is a lack of information on the response of this type of ecosystem to the impact of forest fires, which limits the definition of adequate restoration strategies. Therefore, it is necessary to establish agile and innovative evaluation and monitoring processes; therefore, the objective of this research was to characterize the structure and composition of living fuels and to estimate the average load of dead fuels (woody and litter) in a TCFFW dominated by *P. aquatica* in El Castaño (LEBRE), in a gradient of disturbance (high, medium, low), taking as reference some species diversity parameters. The following hypotheses were put forward: (i) Species diversity is higher in low disturbance sites than in medium and high disturbance sites; therefore, as a consequence of the lower species richness in high disturbance sites, *P. aquatica* acquires a higher value of importance as species diversity decreases. (ii) There are statistical differences in the load of dead fuels (woody and leaf litter) with greater accumulation in sites with high disturbance levels, despite having a lower stratification and species diversity.

## 2. Materials and Methods

### 2.1. Study Area

La Encrucijada Biosphere Reserve (LEBRE) is a natural protected area and is a RAMSAR site [46], located on the southeastern Pacific coast of Mexico. The surface area of LEBRE is 144,868 ha; 24% (36,216 ha) are classified as core areas, while 75.1% (108,651 ha) are classified as buffer areas [47]. *P. aquatica* TCFFWs surround mangroves, freshwater marshes, and other coastal ecosystems (Figure 1), and these ecosystems are connected through a complex hydrological network. In this wetland system, *P. aquatica* is far from the saltwater inlets of the estuaries, and the main source of water is freshwater from the San Nicolas River. The climate is tropical, warm-humid with summer rainfall Am(w). The mean annual temperature is 28 °C, and annual precipitation ranges from 1300 to 3000 mm [48]. The dry period is from February to May; during this period, the water level is below the soil surface [11].

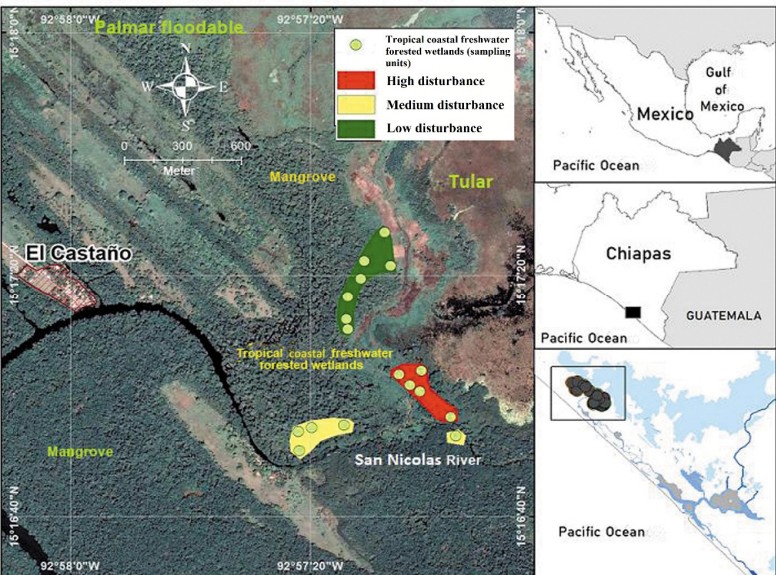

**Figure 1.** Location of the study area in the Encrucijada Biosphere Reserve, México. Legend: The study site is on the Pacific coast of Mexico, and the tropical coastal freshwater forested wetland (TCFFW) polygons show three levels of disturbance.

## 2.2. Field Study

Based on hydroperiod data [11], field campaigns for this study were planned from February to April in 2017 and 2018. Following a stratified randomized experimental design, 17 circular 600 m$^2$ (radius = 13.8 m) sampling units were established in the TCFFW, because the vegetation is dominated by *P. aquatica*; therefore, the study area was considered homogeneous, where the only stratification criterion was the level of disturbance (Figure 1). The disturbance levels in this study were defined taking into account the historical disturbance regime [49] and the intrinsic nature of the disturbances, namely, all those criteria sufficient to define the distribution, frequency, intensity, and rate of return of the disturbance [50]. Disturbance levels were defined based on the following criteria (Table 1): (1) canopy openness, (2) distance from roads, (3) human activities, (4) illegal extraction of wood, (5) hurricane/wind impact, (6) fires, and (7) forest cover. Based on this information, field assessment sites were classified as low disturbance (*n* = 7 sites), medium disturbance (*n* = 5 sites), and high disturbance (*n* = 5 sites) (Figure 2). These sample sizes ensured that the results were statistically significant.

**Table 1.** Site classification criteria of low, medium, and high disturbance.

| Level | Criteria | | | | | | |
|---|---|---|---|---|---|---|---|
| | Canopy Openness | Distance to Roads (km) | Human Activities | Illegal Extraction Wood | Hurricane/Wind Impact | Fires (1998–2018) | Forest Cover (%) |
| Low disturbance | 0 to 25% | 0.6 to 1 | None | Very low | No fallen trees | None | 75–100 |
| Medium disturbance | 25 to 50% | 0.2 to 0.6 | Dirt road | Low to Medium | fallen trees (1 to 3) | Low frequency (5 events) | 50–75 |
| High disturbance | >50% | <0.2 | Dirt road, poaching | High to Very high | fallen trees (>3) | High frequency (>15 events) | <50 |

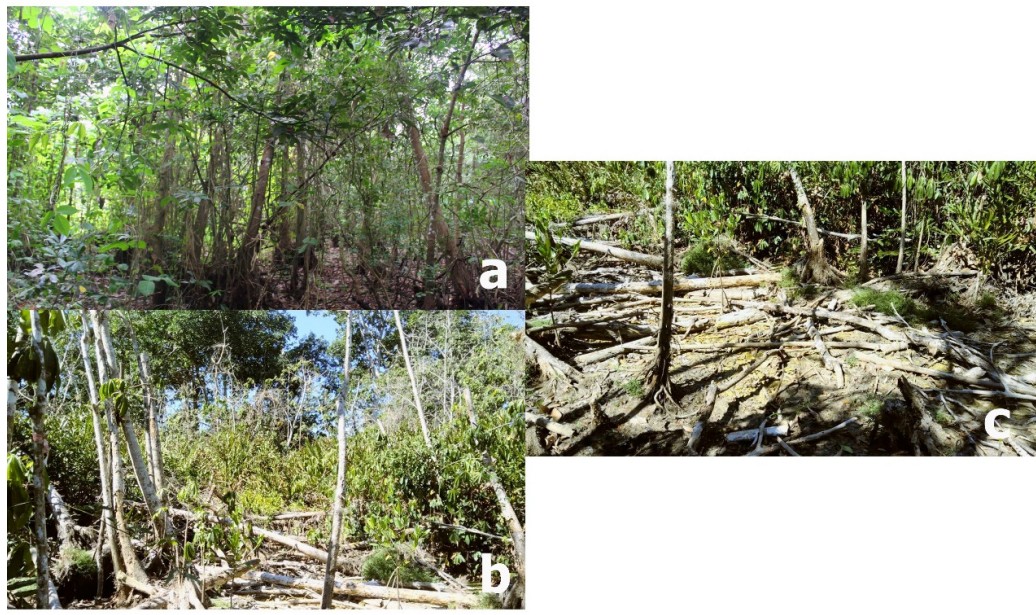

**Figure 2.** Tropical coastal freshwater forested wetlands of *Pachira aquatica*: (**a**) low disturbance, (**b**) medium disturbance, and (**c**) high disturbance.

## 2.3. Structural and Composition Analysis of Fuel Beds

### 2.3.1. Live Fuel

In each sampling unit, measurements of trees, shrubs, and lianas (normal diameter, ND > 2.5 cm) were performed. Height was measured with a hypsometer (Ver-

tex III), and the canopy diameter of 20% of the trees was measured according to [51] (Appendix A). To describe the horizontal structure, 10 diameter classes were established starting at 2.5 cm with 5 cm intervals [52]. Measurements were used to estimate density and basal area [53], relative coverage [54], relative frequency, density, and dominance of species [55] (Appendix A). Furthermore, three structural indices were estimated: the importance value index (IVI) [56], forestry value index (FVI) [54], and Holdridge complexity index (HCI) [57] (Appendix A). The understory was characterized by herbs, shrubs, and trees (height < 50 cm and ND < 2.5 cm) that are part of natural regeneration. To measure this forest stratum, a 60 m$^2$ (diameter = 8.7 m) sub-circle was established in the middle of the SU; records included height, diameter, and coverage.

### 2.3.2. Dead Fuels: Woody Fuel and Litterfall

Woody dead fuel was characterized using the planar intersections technique [58–60], adapted by [45], which is based on quantifying the woody fuels that are intersected by each sampling line that functions as "cutting guillotines". Four 10 m linear transects were established in each sampling unit, from the site center to each cardinal point (N, S, E, W) as shown in Figure 3.

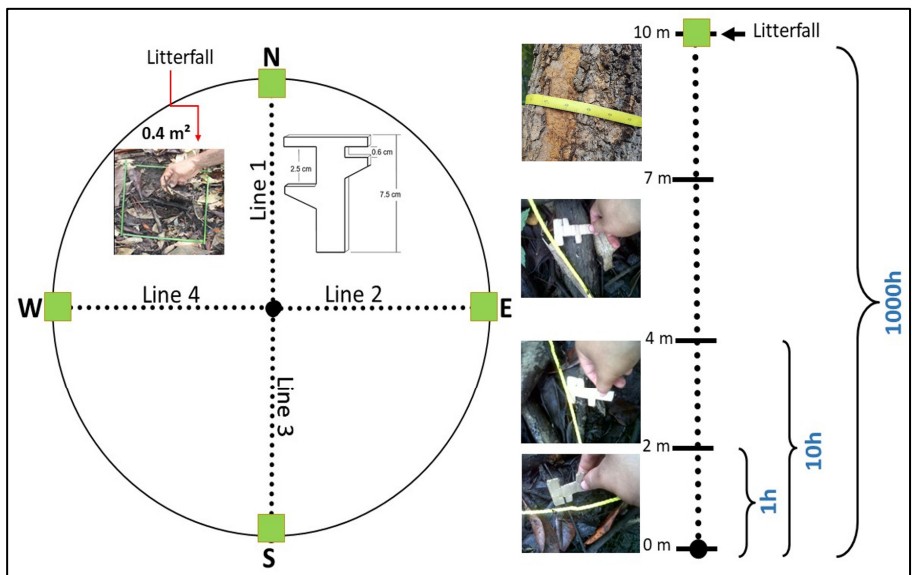

**Figure 3.** Sampling scheme and design of the intersecting lines for the measurement of woody fuels and litterfall. The layout of four line transects, sectioned at different line length intervals to account for the number of intersections for each woody fuel type of woody fuel counted with the use of a metal caliper. Litterfall is assessed at the end of each planar line intersection, with four measurements taken for each sampling unit.

The woody fuel was classified according to the "timelag", i.e., the time necessary for the fuel to lose or gain 66% of its humidity, according to the ambient temperature [24]. Therefore, the timelag/diameter ratio was used (1 h [0–0.6 cm], 10 h [0.61–2.5 cm], 100 h [2.5–7.5 cm], and 1000 h [>7.5 cm]), as detailed in Table 2. Transect measurements were performed as follows: fine fuels (1 h) were evaluated in the first 2 m from the planar intersection line, regular fuels (10 h) were evaluated up to 4 m from the line of planar intersection, the median fuels (100 h) were evaluated up to 7 m, and the fuels of 1000 h were evaluated in the 10 m length of the planar intersection line. Fine fuels (1 and 10 h) and medium fuels (100 h) were counted based on the number of intersections on the horizontal plane (Figure 3). Thick fuels (1000 h) were classified according to their state or condition (firm or rotten), and their diameters were measured.

**Table 2.** Timelag fuel/diameter ratio, fuel class, and equations to estimate woody fuel at each sampling unit at El Castaño according to Brown [58].

| Timelag Fuel | Diameter | Fuel Class | Equation |
|---|---|---|---|
| 1 h | 0–0.6 cm | Fine fuel | $P = (0.484 \times f \times c)/(N \times l)$ |
| 10 h | 0.61–2.5 cm | Regular fuel | $P = (3.369 \times f \times c)/(N \times l)$ |
| 100 h | 2.51–7.5 cm | Medium fuel | $P = (36.808 \times f \times c)/(N \times l)$ |
| 1000 h firm (no rot) | >7.5 cm | Coarse fuel | $P = (1.46 \times d^2 \times c)/(N \times l)$ |
| 1000 h (rotten) | >7.5 cm | Coarse fuel | $P = (1.21 \times d^2 \times c)/(N \times l)$ |

where P = fuel weight (t ha$^{-1}$), $f$ = frequency or number of intersections, $c$ = slope correlation factor, d$^2$ = sum of the squared diameter of branches or logs > 7.5 cm, $N$ = number of intersection lines, $l$ = line longitude or sum of the longitude (2 m = 6.56 feet) (4 m = 13.12 feet) (7 m = 22.96 feet) (10 m = 32.8 feet).

Litterfall was collected at the end of each planar intersection line in four 0.2 × 0.2 m quadrants (Figure 3), where the depth of the layer was measured and separated into surface litterfall and fermented litterfall. Subsequently, litterfall was transferred to the laboratory, and the dry biomass was obtained after 24 h in a drying oven (IKA OVEN) at 105 °C.

2.3.3. Data Processing and Calculations

Average woody fuel beds were estimated following standardized procedures for each diameter class [45,58]. Litterfall dry weight biomass was determined following the methodology of Morfín et al. [61]. For this, it was necessary to obtain the bulk density ($\rho$) at surface litterfall and fermented litterfall depths and subsequently determine the litterfall layer load:

$$\rho = \left( \frac{P}{a * h} \right) \times 10$$

where $\rho$ = bulk density (t ha$^{-1}$ mm$^{-1}$), P = dry weight (g), $a$ = surface area (cm$^2$), $h$ = litterfall depth average (mm), 10 = constant for conversion of litterfall density (g cm$^{-3}$) to (t ha$^{-1}$ mm$^{-1}$). Subsequently, the litterfall f© load (C) was estimated [45,62]:

$$C = \frac{\sum\limits_{i=1}^{8} (hi)(d)}{8}$$

where C = litterfall fuel load (t ha$^{-1}$), $hi$ = litterfall layer (mm) (surface litterfall or fermented litterfall), and $d$ = bulk density (t ha$^{-1}$ mm$^{-1}$).

*2.4. Data Analysis*

Paleontological Statistics software (Past 3.26) [63] was used to analyze diversity, considering the disturbance level for all fuel components (trees, shrubs, herbs, and lianas) studied here. Species richness was estimated using the Margalef index (DMg). At the structural level, Fisher's alpha ($\alpha$) and Shannon–Wiener (H′) indices were used to assess proportional abundance, and Berger–Parker (d) and Simpson (D) indices were used to calculate species dominance for each disturbance level. Differences among disturbance levels were evaluated with an analysis of variance (ANOVA) and Tukey–Kramer test at 95% confidence for comparison of means, using the Statistical Analysis System, version 14 [64]. To define possible differences between dead fuel loads (woody and litterfall), an ANOVA and Tukey–Kramer mean comparison test was also applied, with a significance level of 0.05 using the SAS statistical package.

**3. Results**

*3.1. Structure and Composition of Fuel Beds*

The tree stratum registered eight species in the TCFFW studied at LEBRE, where *P. aquatica* was dominant. The associated species were *Zygia conzattii* (Standl.), *Rhizophora*

*mangle* L., *Lacemoseria racemosa* (L.) C.F. Gaertn., *Cynometra oaxacana* Brandegee, *Leucaena leococephala* (Lam.) de Wit, *Tabebuia rosea* (Bertol.), and *Hampea macrocarpa* Lundell. The shrub stratum had the following composition: *Solanum tampicense* Dunal, and *Malvaviscus arboreus* Cav., Lianas comprised *Paullinia pinnata* L., *Serjania mexicana* (L.) Willd, *Entadopsis polystachya* (L.) Britton, *Mansoa hymenaea* (DC.) A.H. Gentry, *Machaerium kegelii* Meisn, and *Cissus cacuminis* Standl.

In the understory, there were young trees of *P. aquatica* (average height of 0.76 m and average ND of 0.5 m), *R. mangle* (average height of 1.2 m and average ND of 0.7 m), and *L. racemosa* (average height of 1.34 m and average ND of 0.6 m). Other understory species included *C. oaxacana*, *Z. conzattii*, and *L. leucocephala*. The herb stratum mostly comprised *Acrostichum aureum* L. (average height 1.95 m, ND max 1.42 and min 0.98 m) and *Crinum americanum* (average height 0.85 m, ND max 0.66 and min 0.34 m).

### 3.2. Vertical Structure of Fuel Beds

In sites with high disturbance, there was a higher number of trees with heights between 2 and 7 m (75%). In the low and medium disturbance sites, 65% and 49% of trees, respectively, belonged to the 2–7 m category. Therefore, this was the dominant height class in the disturbance gradient (Table 3). The percentage for the 7 to 12 m height class varied across sites, in which only 20, 25, and 32% of the trees were in this category at sites with high, medium, and low disturbance, respectively. Sites with medium disturbance had the highest percentage for classes 12–17 m (14.48%), 17.1–22 m (6.43%), and >22 m (6.43%), compared to no and high disturbance sites (Table 3). Therefore, the sum of the percentages of these three classes corresponds to 24%, while in the other two disturbance conditions (high and null), height classes greater than 12 m represent only 3.29 and 3.16%, respectively.

**Table 3.** Characterization of the vertical structure of living fuels (trees, shrubs, and lianas) in the three levels of disturbance evaluated in tropical coastal freshwater forested wetlands. The species found at each disturbance level, the percentage per species in each height class, and the sum of the percentage of vertical structure represented by each species are shown.

| Species | Height Class (%) | | | | | | Total |
|---|---|---|---|---|---|---|---|
| | >22 m | 17–22 m | 12–17 m | 7–12 m | 2–7 m | <2 m | |
| Low disturbance | | | | | | | |
| *Pachira aquatica* | - | 0.09 | 1.25 | 28.27 | 51.80 | - | 81.41 |
| *Zygia conzattii* | - | - | - | 0.62 | 5.52 | - | 6.14 |
| *Cynometra oaxacana* | - | - | - | 1.07 | 2.14 | - | 3.21 |
| *Entadopsis polystachya* | - | - | - | 0.27 | 2.49 | - | 2.76 |
| *Rhizophora mangle* | - | 0.53 | 0.71 | 0.18 | 0.27 | - | 1.69 |
| *Laguncularia racemosa* | - | 0.09 | 0.53 | 0.27 | 0.62 | - | 1.51 |
| *Hampea macrocarpa* | - | - | - | 0.61 | 0.44 | - | 1.05 |
| *Leucaena leucocephala* | - | - | - | 0.19 | - | - | 0.19 |
| *Paullinia pinnata* | - | - | 0.09 | 0.44 | 0.53 | - | 1.06 |
| *Serjania mexicana* | - | - | - | - | 0.53 | - | 0.53 |
| * Lianas | - | - | - | - | 0.06 | - | 0.06 |
| *Combretum decandrum* | - | - | - | - | 0.03 | - | 0.03 |
| *Tabebuia rosea* | - | - | - | 0.09 | 0.00 | - | 0.09 |
| Total | - | 0.71 | 2.58 | 32.10 | 64.61 | - | 100.00 |
| Medium disturbance | | | | | | | |
| *Pachira aquatica* | 2.39 | 6.06 | 13.94 | 22.21 | 37.25 | 1.83 | 83.68 |
| *Cynometra oaxacana* | - | - | - | 1.65 | 9.17 | - | 11.00 |
| *Zygia conzattii* | - | - | - | 0.55 | 2.02 | - | 2.57 |
| *Entadopsis polystachya* | - | - | - | 0.37 | 0.92 | - | 1.29 |
| *Rhizophora mangle* | 0.73 | 0.37 | 0.18 | - | - | - | 1.28 |
| *Laguncularia racemosa* | - | - | 0.18 | - | - | - | 0.18 |
| Total | 3.12 | 6.43 | 14.48 | 24.78 | 49.36 | 1.83 | 100.00 |

**Table 3.** *Cont.*

| Species | Height Class (%) | | | | | | Total |
|---|---|---|---|---|---|---|---|
| | **>22 m** | **17–22 m** | **12–17 m** | **7–12 m** | **2–7 m** | **<2 m** | |
| High disturbance | | | | | | | |
| *Pachira aquatica* | 0.21 | 0.11 | 0.53 | 18.02 | 70.39 | 0.84 | 90.10 |
| *Laguncularia racemosa* | - | 0.84 | 1.16 | 0.95 | 0.63 | - | 3.58 |
| *Zygia conzattii* | - | - | - | - | 3.06 | - | 3.06 |
| *Rhizophora mangle* | 0.11 | 0.11 | 0.11 | 0.53 | 1.16 | 0.32 | 2.32 |
| * Lianas | 0.00 | 0.00 | 0.00 | 0.32 | 0.42 | 0.00 | 0.74 |
| *Hampea macrocarpa* | 0.00 | 0.00 | 0.00 | 0.00 | 0.21 | 0.00 | 0.21 |
| Total | 0.32 | 1.05 | 1.79 | 19.81 | 75.87 | 1.16 | 100.00 |

* Lianas include: Mansoa hymenaea (DC.) A.H. Gentry, Cissus cacuminis Standl., Machaerium kegelii Meisn, Combretum decandrum Jacq.

### 3.3. Horizontal Structure of Fuel Beds

The analysis of fuel beds based on diameter classes of each disturbance level showed that sites with high disturbance had the highest percentage of ND between 2.5 and 7.5 cm (61.45%). Furthermore, the low disturbance sites had a higher percentage (32.7%) in the 7.5 to 12.5 cm category. However, sites with medium disturbance had the highest percentages in all classes greater than 12.5 cm (Figure 4). Overall, the average percentage for each ND class, including the three disturbance conditions, was as follows: 2.5–7.5 (41.5%), 7.5–12.5 cm (26.2%), 12.5–17.5 cm (18.5%), and 17.5–22.5 cm (8.3%).

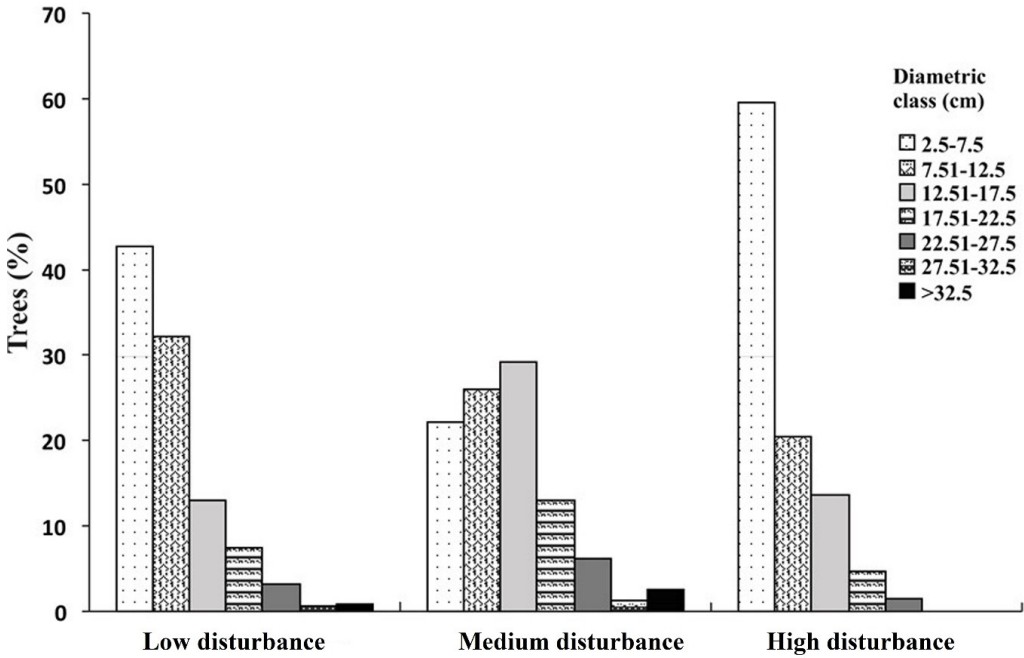

**Figure 4.** Relative histogram by diameter class for each disturbance level.

The average tree density was higher in low disturbance sites, with 2686 individuals ha$^{-1}$ and an average basal area of 26.59 m$^2$ ha$^{-1}$. The highest basal area was observed in sites with medium disturbance (39.41 m$^2$ ha$^{-1}$). The species with the highest IVI was *P. aquatica* (171.35%) at low disturbance sites, followed by *R. mangle* (29.53%) and *Z. conzattii* (24.73%). At sites with medium disturbance, *P. aquatica* (178.07%) had the highest IVI, followed by *C. oaxacana* and *R. mangle*, at 35.36 and 34.44%, respectively. At sites with high disturbance, *P. aquatica* (207.47%) had the highest IVI, while *H. macrocarpa* (4.84%) had the lowest (Table 4). Species composition decreased with disturbance; in this sense, sites with low disturbance had more species (*n* = 12) than sites with high disturbance (*n* = 5).

Regardless of the disturbance level, 33% of the species were present across all sites. In order of importance, these species were *P. aquatica*, *Z. conzattii*, *L. racemosa*, and *R. mangle*, while *H. macrocarpa* was present only at sites with high and low disturbance. *Pachira aquatica* was registered at all three disturbance levels, and it became more dominant as the disturbance increased, which was reflected in its forest value index. The Holdridge index showed that as the number of species decreased, complexity decreased (Table 4).

**Table 4.** Forest fuel bed characterization and structural indices.

| Species | D$_A$ | D$_R$ (%) | F$_A$ | F$_R$ (%) | BA (m$^2$ ha$^{-1}$) | Do$_R$ (%) | IVI (%) | FVI (%) | HCI |
|---|---|---|---|---|---|---|---|---|---|
| **Low disturbance** | | | | | | | | | |
| *Pachira aquatica* | 2171 | 80.85 | 1.17 | 15.56 | 23.72 | 74.95 | 171.35 | 110 | 148.05 |
| *Rhizophora mangle* | 69 | 2.57 | 1.17 | 15.56 | 3.61 | 11.40 | 29.53 | 45 | |
| *Zygia conzattii* | 164 | 6.12 | 1.17 | 15.56 | 0.97 | 3.06 | 24.73 | 53 | |
| *Entadopsis polystachya* | 74 | 2.75 | 1.00 | 13.33 | 0.76 | 2.39 | 18.47 | 25 | |
| *Laguncularia racemosa* | 43 | 1.60 | 0.83 | 11.11 | 0.89 | 2.81 | 15.51 | 23 | |
| *Cynometra oaxacana* | 86 | 3.19 | 0.33 | 4.44 | 1.14 | 3.59 | 11.23 | 19 | |
| *Paullinia pinnata* | 29 | 1.06 | 0.33 | 4.44 | 0.36 | 1.15 | 6.66 | 8 | |
| *Serjania mexicana* | 14 | 0.53 | 0.33 | 4.44 | 0.05 | 0.15 | 5.13 | 5 | |
| * Lianas | 12 | 0.44 | 0.33 | 4.44 | 0.05 | 0.17 | 5.06 | 6 | |
| *Hampea macrocarpa* | 12 | 0.44 | 0.33 | 4.44 | 0.03 | 0.08 | 4.97 | 4 | |
| *Tabebuia rosea* | 2 | 0.09 | 0.17 | 2.22 | 0.06 | 0.20 | 2.52 | - | |
| *Leucaena leucocephala* | 7 | 0.27 | 0.17 | 2.22 | 0.01 | 0.03 | 2.52 | 2 | |
| *Combretum decandrum* | 2 | 0.09 | 0.17 | 2.22 | 0.01 | 0.02 | 2.33 | - | |
| Total | 2686 | 100 | 7.5 | 100 | 26.59 | 100 | 300 | 300 | |
| **Medium disturbance** | | | | | | | | | |
| *Pachira aquatica* | 1900 | 81.72 | 1.00 | 21.05 | 29.67 | 75.30 | 178.07 | 146 | 102.59 |
| *Cynometra oaxacana* | 250 | 10.75 | 1.00 | 21.05 | 1.40 | 3.55 | 35.36 | 77 | |
| *Rhizophora mangle* | 29 | 1.25 | 0.75 | 15.79 | 6.86 | 17.40 | 34.44 | 29 | |
| Lianas | 54 | 2.33 | 0.75 | 15.79 | 0.30 | 0.75 | 18.87 | 12 | |
| *Zygia conzattii* | 58 | 2.51 | 0.75 | 15.79 | 0.22 | 0.55 | 18.85 | 21 | |
| *Entadopsis polystachya* | 29 | 1.25 | 0.25 | 5.26 | 0.85 | 2.15 | 8.67 | 9 | |
| *Laguncularia racemosa* | 4 | 0.18 | 0.25 | 5.26 | 0.12 | 0.30 | 5.74 | 6 | |
| Total | 2325 | 100 | 4.75 | 100 | 39.41 | 100 | 300 | 300 | |
| **High disturbance** | | | | | | | | | |
| *Pachira aquatica* | 2386 | 90.61 | 1.00 | 27.27 | 25.60 | 89.59 | 207.47 | 195 | 72.03 |
| *Rhizophora mangle* | 53 | 2.00 | 0.83 | 22.73 | 1.74 | 6.08 | 30.81 | 29 | |
| *Laguncularia racemosa* | 89 | 3.38 | 0.67 | 18.18 | 0.65 | 2.26 | 23.82 | 37 | |
| *Zygia conzattii* | 81 | 3.06 | 0.67 | 18.18 | 0.51 | 1.78 | 23.02 | 25 | |
| Lianas | 19 | 0.74 | 0.33 | 9.09 | 0.06 | 0.21 | 10.04 | 11 | |
| *Hampea macrocarpa* | 6 | 0.21 | 0.17 | 4.55 | 0.02 | 0.08 | 4.84 | 3 | |
| Total | 2633 | 100 | 3.67 | 100 | 28.57 | 100 | 300 | 300 | |

Absolute (DA) and relative (DR) densities, absolute (FA) and relative (FR) frequencies, basal area (BA), importance value index (IVI), forest value index (FVI), and Holdridge complexity index (HCI). * Lianas include: *Mansoa hymenaea*, *Cissus cacuminis*, *Machaerium kegelii*, *Combretum decandrum*.

Species richness was highest for low disturbance sites (19 species), followed by medium disturbance (14 species) and high disturbance (9 species) sites. Alpha Fisher index, Margalef, Shannon, and Simpson were all higher in non-disturbed sites. The Berger–Parker index was higher for *P. aquatica* in highly disturbed sites (Table 5).

**Table 5.** Fuel beds, live-fuel characterization.

| FB | S | α | DMg | H′ | d | D |
|---|---|---|---|---|---|---|
| Low disturbance | 19 | 2.91 ± 0.75 | 2.39 ± 0.41 | 1.06 ± 0.01 | 0.77 ± 0.003 | 0.39 ± 0.02 |
| Medium disturbance | 14 | 1.8 ± 0.3 | 1.54 ± 0.19 | 0.91 ± 0.11 | 0.78 ± 0.04 | 0.37 ± 0.007 |
| High disturbance | 9 | 1.16 ± 0.08 | 1.01 ± 0.58 | 0.56 ± 0.22 | 0.87 ± 0.02 | 0.22 ± 0.05 |

Average and standard deviation of each index. S = richness; α = Alpha Fisher; DMg = Margalef; H′ = Shannon–Wiener; d = Berger–Parker; D = Simpson.

*3.4. Dead Fuels*

The woody fuel stratum of fuel beds in TCFFW was different among the 1, 10, and 1000 h (rotten) classes. The highest fuel load was observed at sites with high disturbance, while in the remaining categories, there were no significant differences. However, litterfall fuels (surface litterfall and fermented litterfall) did not show differences in the disturbance gradient studied (Table 6). The results of the total woody fuel load (including all classes) showed that the highest average amount was $176.22 \pm 31.48$ t ha$^{-1}$ (high disturbance), followed by medium disturbance ($130.15 \pm 21.75$ t ha$^{-1}$) and low disturbance ($121.36 \pm 27.15$ t ha$^{-1}$). The average total litterfall fuel load (which included the two evaluated categories) was similar among high disturbance ($45.96 \pm 8.42$ t ha$^{-1}$), medium disturbance ($46.51 \pm 5.11$ t ha$^{-1}$), and low disturbance ($43.83 \pm 3.86$ t ha$^{-1}$) sites.

**Table 6.** Average value of woody fuels by diameter class and litterfall.

| **Tropical Coastal Freshwater Forested Wetlands/Disturbance Level** | | | | |
|---|---|---|---|---|
| **Level** | **Fuel Beds Stratum Average (t ha$^{-1}$)** | **SD** | | |
| 0–0.6 cm (1 h) | | | | |
| Low disturbance | 4.33 | 1.21 | ab | $F_{2,14} = 4.94$ |
| Medium disturbance | 3.26 | 0.23 | b | $p = 0.02$ |
| High disturbance | 5.93 | 0.65 | a | |
| 0.61–2.5 cm (10 h) | | | | |
| Low disturbance | 10.72 | 0.93 | b | $F_{2,14} = 19.92$ |
| Medium disturbance | 11.82 | 1.21 | b | $p \leq 0.0001$ |
| High disturbance | 15 | 1.54 | a | |
| 2.51–7.5 cm (100 h) | | | | |
| Low disturbance | 42.86 | 5.65 | | $F_{2,14} = 1.45$ |
| Medium disturbance | 51.28 | 11.95 | | $p = 0.26$ |
| High disturbance | 55.55 | 19.9 | | |
| >7.5 cm rotten (1000 h) | | | | |
| Low disturbance | 48.7 | 26.89 | b | $F_{2,14} = 8.47$ |
| Medium disturbance | 47.08 | 13.42 | b | $p = 0.0039$ |
| High disturbance | 90.34 | 12.98 | a | |
| >7.5 firm (1000 h) | | | | |
| Low disturbance | 14.42 | 9.33 | | $F_{2,14} = 1.57$ |
| Medium disturbance | 15.71 | 3.17 | n.s. | $p = 0.24$ |
| High disturbance | 11.55 | 2.86 | | |
| Surface litterfall | | | | |
| Low disturbance | 27.85 | 3.47 | | $F_{2,14} = 0.13$ |
| Medium disturbance | 28.42 | 4.59 | n.s. | $p = 0.87$ |
| High disturbance | 25.66 | 3.73 | | |
| Fermentation litterfall | | | | |
| Low disturbance | 15.97 | 8.54 | | $F_{2,14} = 0.50$ |
| Medium disturbance | 18.08 | 7.81 | n.s. | $p = 0.61$ |
| High disturbance | 20.3 | 6.54 | | |

Significant differences are shown with different letters, as determined by the Tukey–Kramer test ($p < 0.05$). ($\pm$SD = standard deviation). n.s. = not significant.

Finally, the total amount of dead forest fuels (woody and litterfall) was significantly higher in sites with high disturbance ($222.19 \pm 13.37$ t ha$^{-1}$), while the lowest load corresponded to sites with low disturbance ($165.2 \pm 34.92$ t ha$^{-1}$) (Figure 5).

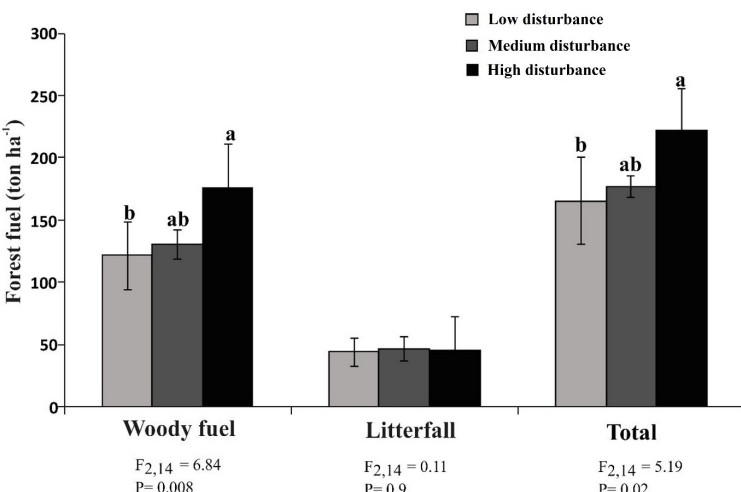

**Figure 5.** Woody fuel, litterfall, and total forest fuel load. Legend: Significant differences are shown with different letters, as determined by the Tukey–Kramer test ($p < 0.05$), ($\pm$standard deviation).

## 4. Discussion

Despite the controversy about how disturbance influences diversity [65], there is some evidence that less disturbance leads to higher species diversity [66]. Tropical coastal freshwater forested wetlands in the southeastern Mexican Pacific are sensitive to disturbance. In this study, we observed that sites with medium to low disturbance had higher diversity and were richer with up to 12 species. However, the number of species was low compared to other forested tropical freshwater coastal wetlands in the Gulf of Mexico [67–70]. For instance, in Calakmul, Campeche, the number of species can be as high as 56 [71]. It is possible that the lower diversity in El Castaño (Pacific) is related to the high dominance of *P. aquatica* [11]. In this region, *P. aquatica* is a very adaptive species that grows in association with some mangrove species [45].

The dominance of *P. aquatica* was potentially associated with its germination requirements, which can have up to 18 practical salinity units (SPUs) in the Pacific, in contrast to the Gulf of Mexico [11]. The surface of TCFFW in El Castaño and the region has decreased in recent years due to anthropogenic activities, such as tree harvesting, agriculture, livestock, and villages [8,72]. This pressure on the ecosystem could lead to a successional stage, where the tree stratum cover decreases while the shrub and herb stratum may be more abundant [73]. From this perspective, the characterization of the structure and composition of forest fuel beds are of great interest and should be studied in the long term. Precisely, an important field of application for studies related to the structural composition of forests is fire ecology, mainly in the TCFFW. These ecosystems have been affected by an irresponsible fire use regime [74] and fire dynamics have also contributed to changes in their vegetation [75–78], thus reducing their diversity.

In this study, the Shannon–Wiener (H') indices in undisturbed forest areas were similar to those of other TCFFWs [67,79,80], while the values obtained in high disturbance regions were low, as expected. In the Gulf of Mexico, the *P. aquatica* TCFFW can be low (4 m) and grow up to 30 m, as indicated by Moreno and Infante [5]. In contrast, in the Pacific, the same type of species grows as much as 17 m [11], values similar to those reported in this study. Furthermore, 59.1% of the canopy strata were from 6 to 17 m, while smaller trees represented only 6%. However, this was not the case in high disturbance areas, where the most abundant height (76%) ranged from 2 to 7 m, thus verifying that disturbance had an effect on the vertical structure.

Medium disturbance in the Pacific TCFFW caused *P. aquatica* to grow higher than in undisturbed sites. This suggests that low fire frequency could be beneficial to the ecosystem [81,82], acting as a renewal mechanism. In areas with high disturbance, *P. aquatica* was more dominant but did not have a great height. Therefore, the level of disturbance influenced the vertical and horizontal structure, and in sites with a higher

density of *P. aquatica*, the forest will not be as tall. The most common diameter class of live fuels on the horizontal plane ranged from 2.5 to 7.5 cm, independent of disturbance. Similar observations have been reported at other sites of the LEBRE, e.g., El Jicaro [11]. Similarly, in TCFFW from the Gulf of Mexico in Veracruz, the most frequent class was from 3 to 13 cm [2]. This was especially observed in El Castaño in high disturbance sites. The density of trees made up of the live fuel stratum was higher in this study and had a similar range to that reported in sites in the Gulf of Mexico [7]. However, in other LEBRE sites with a higher state of conservation, such as Brisas del Hueyate, the density can reach up to 3310 ind. ha$^{-1}$ and the IVI of *P. aquatica* (285.28%) was higher [12].

In the present study, the density of trees comprising the live fuel's stratum was higher in the less disturbed sites, while woody and litter fuels showed the opposite pattern, having more accumulation in sites with greater disturbance. However, the average value of dead fuels (woody and litterfall) reveals that TCFFWs from the Pacific coast can have more fuel than other forested ecosystems in the tropics [44,83] and temperate regions [84–86]. In the study carried out by Barrios-Calderón et al. [45] in the Brisas del Hueyate area within the LEBRE, loads were reported for the classes of 1 (2.75 $\pm$ 0.45 t ha$^{-1}$), 10 (7.01 $\pm$ 1.65 t ha$^{-1}$), and 100 h (18.58 $\pm$ 7.22 t ha$^{-1}$) that are less than those obtained in this study. However, for the class of 1000 h rotten (20.67 $\pm$ 16.22 t ha$^{-1}$) and firm (14.18 $\pm$ 9.33 t ha$^{-1}$), these same authors reported loads similar to those obtained at low disturbance TCFFW sites in the present study. Thus, TCFFWs with medium and high disturbance represent a greater potential to present fires under favorable temperature and weather conditions. Mainly high disturbance sites would represent hot spots for fires with the worst implications for the ecosystem [87].

Litterfall fuels, especially the surface litterfall (SL) from all studied sites, are higher than those reported by Rodríguez et al. [42] in Quintana Roo, Campeche, and Yucatan, with loads up to 17.2 t ha$^{-1}$. Litterfall productivity in TCFFW from the Gulf of Mexico (Veracruz) ranged from 9.3 $\pm$ 0.5 t ha$^{-1}$ (Apompal) to 14.9 $\pm$ 1.0 t ha$^{-1}$ (Chica) [10], which is similar to that observed in this study. In both the Gulf of Mexico and the southeast Pacific (LEBRE, El Castaño), litterfall fuels are the result of tree and liana productivity. As suggested by Souza et al. [88], high litterfall accumulation is determined by climatic factors that affect the vegetative phenology of tree species. Thus, the amount of litterfall fuels in the soil of the study area is twice that accumulated in Veracruz TCFFWs in the Gulf of México; thus, the accumulation rate is higher in the southeastern Pacific. In terms of fire potential, this also represents a greater threat for the El Castaño area, due to the high accumulation of these litterfall fuels [89,90]. However, the litterfall fuels in the fermented layer are lower compared to studies in the Yucatan Peninsula [42], with maximum loads of 53.89 t ha$^{-1}$. This layer is especially important because combustion is low, but energy is high; therefore, underground fires are more severe than surface fires [91]. Observations of this fuel layer in this study show no significant differences in the three conditions evaluated; therefore, regardless of the disturbance level, TCFFWs are vulnerable to underground fires in the LEBRE.

Some authors, such as Rodriguez [92] point out that the amount of woody fuels and litterfall decreases with frequent fires. However, despite the fact that, in the last 10 years, two fires of low intensity have been recorded in the study area [31], a limited decrease in dead fuels has been observed in sites with high or medium disturbance. This is due to disturbances caused by timber extraction and the opening of new roads, according to information provided by forest rangers. Regarding the average load of dead fuels (woody and litterfall), Barrios-Calderón et al. [45] observed an average load of 225.06 t ha$^{-1}$ at other study sites in the TCFFW of El Castaño. These loadings are very similar to those of high disturbance sites, which represent the highest average accumulation of dead fuels. In another study carried out at the TCFFW in Calakmul, Campeche, Contreras et al. [93] reported total loads ranging from 43.15 to 154.5 t ha$^{-1}$, which were within the range obtained for no and medium disturbance sites. However, the total amount of dead fuels was higher than other TCFFW dominated by *P. aquatica* in the Yucatan Peninsula, Mexico

(16.46 t ha$^{-1}$), as shown by Reyes and Coli [83], as they are the lowest loads that could be considered in relation to those obtained in this work.

Recently, Flores et al. [60] determined that the load or biomass of these dead fuels contains between 47.5% C for litter and 67% C for 1000 h firm fuels. Consequently, the total amount of C released into the atmosphere when the TCFFW is burned contributes to the amount of greenhouse gases. Thus, the conservation and management of these coastal ecosystems are important for the C cycle at the local and regional scales. Therefore, Mahdizadeh and Russell [94] pointed out the need not only to estimate degraded areas, but also to quantify the amount of C lost in the most disturbed areas in a given year, which can vary greatly.

Irrational logging causes alteration in the vegetation structure, which has led to a greater increase and continuity of forest fuels, resulting in a greater possibility of fires [95]. Fuel beds in highly disturbed sites have a higher ignition potential and C emissions through fine fuels (1 and 10 h) and litterfall, while the medium fuels (100 h) spread the fire; coarse fuels (1000 h) are related to their intensity. Furthermore, dead fuel available at any level of forest disturbance is related to the potential for fire spread and movement on stairway fuels [96]. The latter is the result of the vertical continuity of fuels arranged in these ecosystems, ranging from herbs, grasses, shrubs, lianas, and trees connecting one stratum with the next. In general, sites with higher disturbances will have higher fire potential; these sites will require the implementation of strategies to prevent and mitigate fire. Furthermore, if there is a wide dominance of *P. aquatica*, a type of softwood that grows in tropical regions, conditions increase the decomposition rates of this type of woody material [97]. Finally, the probabilities of underground fires are similar in the three TCFFW conditions, as there are no differences in the depth of the litterfall layer or in the amount of organic material.

The information generated in this study related to the characterization of fuel beds is a starting point for further studies to predict the ignition, spread, and impact of fires in these ecosystems. In general, the potential for fires in the TCFFW conditions evaluated is more evident in areas with high disturbance that require the implementation of management and conservation strategies.

Understanding these aspects is relevant for supporting or implementing restoration strategies for forest ecosystems, for which continuous evaluations (monitoring) should be carried out to show whether succession dynamics are tending towards restoration of these ecosystems. In this way, it will be possible to determine which aspects favor and which hinder restoration processes at different stages of development [98] (Navarro-Martínez et al., 2012). Furthermore, it is important to consider that areas affected by fires can be used not only by native vegetation, but also by exotic species [99]; therefore, a rapid detection of this scenario will help to make the trend of the restoration process more effective.

## 5. Conclusions

Observations show that in low disturbance sites, greater species diversity can be found, while in medium and high disturbance sites, richness decreases; therefore, although the species *P. aquatica* has a high importance value that makes it a dominant species under disturbed conditions, its vertical and horizontal structures do not show optimal development. Large amounts of woody fuels lead to a higher probability of occurrence of high-to-severe fires. The composition of fuel beds in the high and middle disturbance forests showed a greater accumulation of forest fuels. The proximity to the population center, the opening of roads or access routes, the illegal extraction of trees and shrubs, and the historical record of fires have become causes of the high accumulation of forest fuels. Therefore, it is important to consider that the probability of the occurrence of surface fires increases. However, the three TCFFW evaluated conditions have the same probability of presenting underground fires, where the intensity depends on the depth of the organic material (fermented litterfall) that is equally distributed in the three disturbance conditions evaluated. TCFFWs are an important carbon sink, which, during a fire event, could release considerable amounts of greenhouse gases into the atmosphere. The information from

this study helps to define and prioritize areas that need different management strategies in these ecosystems, for which there are few references. It is recommended in future research to consider an analysis of indicator species to characterize those species with fidelity of occurrence (preference) in sites with low, medium, and high disturbance, as well as a management analysis to represent the distribution of the species in the different disturbance levels. This would allow for a better understanding of the quantity, quality, and spatial distribution of dead forest fuels with respect to the structure and composition of species and their contribution in terms of biomass within the studied ecosystem. It is recommended that the results of this research be used to focus forest fuel management planning efforts in TCFFW at regional and national levels towards the creation of fuel management regimes that increase social and ecological resilience to wildfire.

**Author Contributions:** Conceptualization, R.d.J.B.-C., D.I.M. and J.G.F.G.; methodology, R.d.J.B.-C., D.I.M. and J.G.F.G.; formal analysis and investigation, R.d.J.B.-C., D.I.M., J.G.F.G. and J.R.T.; writing—original draft preparation, R.d.J.B.-C., D.I.M. and J.G.F.G.; writing—review and editing, R.d.J.B.-C., D.I.M. and J.R.T.; funding acquisition, D.I.M. All authors have read and agreed to the published version of the manuscript.

**Funding:** This research received no external funding.

**Data Availability Statement:** The datasets used and/or analyzed during the current study are available from the corresponding author upon reasonable request.

**Acknowledgments:** We would like to thank El Colegio de la Frontera Sur (ECOSUR), the Consejo Nacional de Ciencia y Tecnología (CONACYT) for the awarded scholarship (No. 429018), and the Comisión Nacional de Áreas Naturales Protegidas (CONANP) for the facilities provided to carry out this work.

**Conflicts of Interest:** The authors declare no conflicts of interest.

## Appendix A

**Table A1.** Formulas used to describe structural parameters in the live fuel stratum of tropical coastal freshwater forested wetlands.

| Variable | Formula | Variable Description | Reference |
|---|---|---|---|
| Basal area | $BA = (\pi \times 4\,([ND])^2)$ | $BA$ = total basal area in $m^2/0.1$ ha<br>$\pi = 3.1416$<br>$ND$ = normal diameter | Ramos et al. [50] |
| Absolute/Relative Coverage | $C_A = (((d_1 + d_2)/2)/2)^2 \times \pi$<br>$C_R = (Cov_i/Cov_t) \times 100$ | $C_A$ = absolute coverage<br>$d_1$ and $d_2$ = canopy diameters<br>$\pi = 3.1416$<br>$C_R$ = relative coverage<br>$Cov_i$ = absolute coverage of individuals of a species<br>$Cov_t$ = absolute coverage of individuals of all species | Zarco et al. [51] |
| Relative Frequency | $F_R = f_i/N \times 100$ | $F_R$ = relative frequency<br>$f_i$ = number of occurrences of a species<br>$N$ = number of occurrences of all species | Gentry and Ortiz [55] |
| Relative Density | $D_R = n_i/N \times 100$ | $D_R$ = relative density<br>$n_i$ = number of individuals of a species<br>$N$ = number of individuals of all species at each sampling site | Villavicencio and Valdez [56] |
| Dominance Relative | $Do_R = (\alpha_i/\alpha) \times 100$ | $\alpha_i$ = basal area of a species at each sampling site<br>$\alpha$ = total basal area of all species at each sampling site | Villavicencio and Valdez [56] |
| Importance Value Index | $IVI = D_R + Do_R + F_R$ | $D_R$ = relative density<br>$Do_R$ = relative dominance<br>$F_R$ = relative frequency | Villavicencio and Valdez [56] |
| Forest Value Index | $FVI = ND_R + H_R + C_R$ | $ND_R$ = normal diameter relative<br>(ND absolute of a species/ND absolute of all species)<br>$H_R$ = relative height<br>(absolute height of a species between the absolute height of all species)<br>$C_R$ = (canopy diameter of all species/sampled area) × 100 | Zarco et al. [54] |
| Holdridge Complexity Index | $HCI = (d \times a \times h \times s)/100$ | $d$ = number of trees in the sample unit/0.1 ha<br>$a$ = total basal area ($m^2/0.1$ ha). It was calculated with the formula $\pi \times 4(DAP)^2$<br>$h$ = average stand height in meters<br>$s$ = total number of species in the sample unit (0.06 ha) | Holdridge et al. [57] |

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
