# Peer review of "Forest Fuel Bed Variation in Tropical Coastal Freshwater Forested Wetlands Disturbed by Fire"

_forests, doi:10.3390/f15010158_

Round 1

Reviewer 1 Report

Comments and Suggestions for Authors The authors investigated and analyzed the structure and composition of fuel beds in tropical coastal freshwater forested wetlands undter different disturbance conditions. The results may provide some basic information for the region and provide reference for forest fire prevention, etc. However, there are some issues require further clarification:   (1)            The regional representativeness of the sampling units was not specified in detail. Can these 17 sampling units represent the main forest types in the region? (2)            What is the basis for judging gradient of disturbance (high, medium, low)? (3)            It is recommended to increase some sampling points to enrich the data base, in order to further reveal the relationships the relationship between disturbance and diversity. (4)            The current work of the manuscript is mainly about field sampling statistics, and the innovation seems to be insufficient.

Reviewer 2 Report

Comments and Suggestions for Authors

This paper characterized the structure and composition of fuel beds in tropical coastal freshwater forested wetlands (TCFFW) with three levels of disturbance at El Castaño, La En-crucijada Biosphere Reserve. Especially, the authors investigated the influence of disturbance levels on dead fuel load of the TCFFW. This study contributes to defining areas prone to fire in these ecosystems and designing prevention strategies. The results are novel, interesting, and publishable. But I feel that the authors still have some work to do before the manuscript can be published.

Although the article is clear and fluent, it is overwritten in some places, such as the results, discussion, and conclusion. I suggest that the author condense the core result and build a discussion and summary around it.

1. The title of the paper is too long. It is suggested to further condense the title from the core findings of the study.

 2. What are forest fuel beds, although covered in the introduction, is relatively general and is suggested to be clarified briefly in the introduction.

 3. The normal diameter of a tree should be measured using a diameter gauge, Figure 3 shows that the diameter of the tree is measured with a steel tape. Although the diameter of the tree can be obtained by this method, the error is large.

 4. Senction3.1. Structure and composition of fuel beds. It is recommended to make a table to see the parameters more clearly.

Reviewer 3 Report

Comments and Suggestions for Authors

Review the relevant literature to clarify the criteria used to define three disturbance levels in coastal wetlands at an international scale

Provide explicit details on the methodology for categorizing disturbance levels

Highlight variability within and between sampling units

Discuss specific factors contributing to fire vulnerability, linking ecological, climatic, or anthropogenic variables for practical relevance

Integrate study findings into actionable recommendations for ecosystem management and fire prevention, bridging scientific knowledge with real-world conservation applications

Comments on the Quality of English Language

The quality of the English language in this document is commendable. But improve some typos and local terms. 

Round 2

Reviewer 1 Report

Comments and Suggestions for Authors

After revision, I think the manuscript can be considered for acceptance.